# EquiCAD: A Geometric Equivariant Neural Network for 3D Shape Classification

## Abstract

Three-dimensional (3D) shape classification plays a central role in computer vision and computer-aided design (CAD), underpinning applications in intelligent manufacturing, automated inspection, and digital engineering. Despite recent progress with 3D CNNs and graph-based approaches, existing methods often overlook the geometric-topological regularities and symmetry principles intrinsic to CAD boundary representations (B-reps). To address this challenge, we introduce EquiCAD, a symmetry-aware learning framework that integrates equivariant representations with graph-based reasoning. By leveraging group-theoretic decomposition of curve and surface descriptors, EquiCAD enforces consistent $SO(3)/O(3)$-equivariance while preserving rich geometric details. The model further exploits hierarchical message passing to capture interactions between local features and global structure. Experimental results across multiple datasets, including SolidLetters, Parts, and a newly constructed benchmark, demonstrate substantial improvements over prior state-of-the-art approaches, particularly on industrially relevant shapes with fine-grained attributes. These findings highlight the value of symmetry-aware modeling for robust and generalizable 3D shape analysis.

## 1 Introduction

Three-dimensional shape classification is a cornerstone problem in computer vision and geometric deep learning, with critical applications ranging from autonomous driving and robotic manipulation to industrial design and digital content creation (Park et al., 2019; Wu et al., 2015; Xu et al., 2023). Among various 3D representations, Boundary Representation (B-rep) CAD models (Jayaraman et al., 2021b; Zhou et al., 2021; Zhang et al., 2024) are particularly significant in manufacturing and engineering, as they offer a precise, compact, and feature-rich description of solid objects. However, the complex, structured nature of B-rep data—comprising vertices, edges, and faces along with their intricate topological relationships—presents a formidable challenge for automated recognition systems. The task becomes especially difficult when aiming for fine-grained geometric perception that is not only accurate but also robust to rigid transformations such as rotations and reflections, a fundamental requirement for real-world deployment (Thomas et al., 2018; Fuchs et al., 2020; Geiger et al., 2022).

Early attempts at CAD shape analysis often relied on handcrafted, rule-based methods that leverage domain-specific geometric heuristics. While interpretable, these approaches are inherently limited in scalability and generalization, as they require extensive manual feature engineering and struggle to adapt to complex or unseen shape variations. The advent of data-driven techniques initially favored point-based or voxelized representations due to their compatibility with standard neural architectures (Park et al., 2019; Mildenhall et al., 2020). However, such methods discard the rich topological structure and parametric geometry of B-reps, leading to suboptimal performance on fine-grained, feature-aware tasks. Moreover, they are often sensitive to sampling density and fail to preserve continuity and symmetry properties, limiting their applicability to high-precision CAD analysis.

More recent graph-based methods (Jayaraman et al., 2021b; Wu et al., 2024; Li et al., 2025) have sought to model B-reps by representing topological entities as nodes and their adjacencies as edges. Although these approaches better preserve structural information, they typically overlook the fundamental role of symmetry. Standard graph neural networks (GNNs) applied to B-rep graphs are

not inherently equivariant to 3D rotations and reflections, causing them to learn pose-dependent features that hamper robustness and generalization (Bronstein et al., 2017; Cohen & Welling, 2017). Furthermore, many GNN-based models do not explicitly account for the heterogeneous nature of B-rep entities (e.g., faces, edges, loops) or the multi-level interactions between local geometric descriptors and global shape characteristics. This often results in limited expressiveness for capturing hierarchical part-whole relationships and symmetry-aware shape semantics.

To overcome these limitations, we introduce EquiCAD, a symmetry-aware learning framework that integrates group-theoretic equivariance with hierarchical graph-based reasoning for B-rep classification. Our method is the first to systematically unify $SO(3)/O(3)$-equivariant neural networks with graph representation learning on CAD data (Cohen & Welling, 2017; Thomas et al., 2018). EquiCAD introduces a symmetry-preserving encoding scheme that decomposes geometric entities from B-reps into irreducible representations based on physical parity, thereby guaranteeing rigorous $SO(3)/O(3)$-equivariance throughout the model. Additionally, the framework employs hierarchical message passing to capture multi-scale interactions between local features and global symmetry constraints. Extensive experiments on three benchmarking datasets demonstrate that EquiCAD achieves state-of-the-art performance, underscoring the critical role of symmetry-aware modeling in advancing 3D shape understanding.

Our main contributions are summarized as follows:

- We introduce EquiCAD, the *first* framework that systematically unifies $SO(3)/O(3)$-equivariant neural networks with graph neural networks for 3D solid model classification.

- We propose a symmetry-preserving encoding scheme that rigorously decomposes B-rep entities into irreducible representations based on their physical parity, thereby guaranteeing end-to-end $SO(3)/O(3)$-equivariance.

- We curate and release a new benchmarking dataset, featuring 16 challenging sub-tasks for fine-grained feature recognition in CAD models, to facilitate rigorous evaluation of equivariance-based methods.

- Through extensive experiments on three benchmarks, we demonstrate that EquiCAD significantly outperforms state-of-the-art baselines, validating the importance of symmetry priors for robust and generalizable 3D shape classification.

## 2 RELATED WORK

**3D Shape Classification with Deep Learning.** The advent of deep learning has spurred diverse approaches to 3D shape understanding. Early methods based on voxelized grids or multi-view images (Wu et al., 2015; Park et al., 2019; Mildenhall et al., 2020) often struggled with computational inefficiency or loss of 3D geometric information. Point-based architectures, such as PointNet(Qi et al., 2017a) and its successors (Qi et al., 2017b; Wang et al., 2019), revolutionized the field by directly consuming unordered point clouds. While these methods achieve permutation invariance, they are inherently designed for unstructured data and do not capitalize on the rich, structured geometric and topological information inherent to CAD boundary representations (B-reps) (Guo et al., 2020; Bronstein et al., 2017), which is critical for industrial applications.

**Neural Networks for CAD Data.** To better exploit the structure of CAD models, recent research has focused on representations that natively encode B-rep data. UV-Net (Jayaraman et al., 2021a) pioneered a hybrid architecture that processes individual B-rep faces using 2D CNNs on their parameter space (UV patches) and models their topological connectivity via a graph neural network (GNN). This approach demonstrated superior performance on CAD-specific tasks compared to point- or voxel-based methods. Subsequent works (Wu et al., 2021; Zhou et al., 2021; Zhang et al., 2024) have explored hierarchical GNNs and attention mechanisms to capture multi-scale relationships in the B-rep graph.

**$SO(3)/O(3)$-Equivariant Deep Learning.** Equivariant neural networks provide a principled framework for building invariance or equivariance to specific symmetry groups (e.g., $E(3)$, $SO(3)$, $O(3)$) directly into the model architecture (Fuchs et al., 2020; Schütt et al., 2017; Weiler et al., 2018). By leveraging steerable features and irreducible representations (irreps), models such as those built

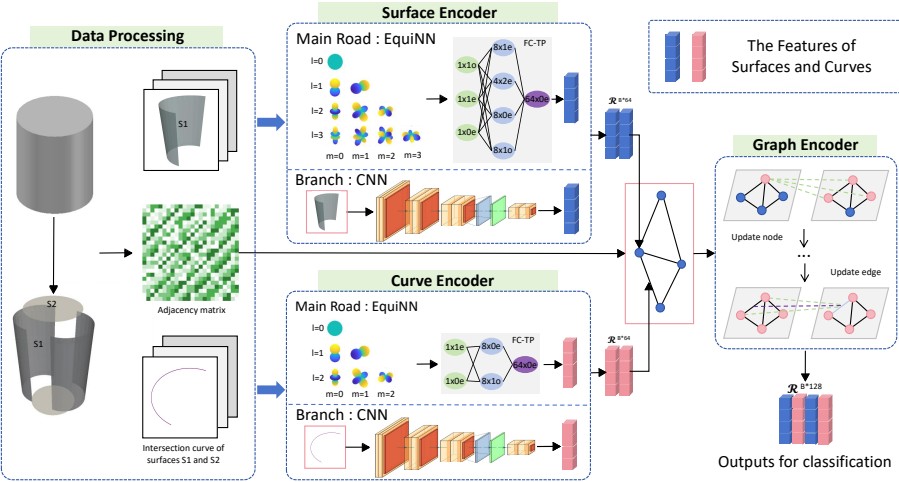

Figure 1: Overview of the proposed EquiCAD framework. Our approach extracts meaningful features of B-rep models through three synergistic steps: (1) an $SO(3)/O(3)$-equivariant encoder that preserves geometric symmetries, (2) a conventional CNN branch that captures local UV-space patterns, and (3) a graph neural network that aggregates topological relationships. Feature fusion combines the strengths of both encoding strategies before message passing produces the final graph-level embedding for classification.

with the e3nn framework (Geiger et al., 2022) have shown remarkable data efficiency and generalization in physical and geometric domains. However, the application of these powerful principles has been largely confined to relatively simple 3D data types like point clouds, spherical signals, or voxels.

**Hybrid and Multi-Modal Architectures.** Bridging different neural network paradigms has proven effective for tackling complex data. Hybrid models that combine CNNs with GNNs (Wu et al., 2024; Li et al., 2025; Hanocka et al., 2019) can capture both local patterns and relational structure. Similarly, recent efforts have begun to integrate equivariant layers with standard architectures (Geiger et al., 2022; Thomas et al., 2018; Fuchs et al., 2020) to balance symmetry-awareness with high-dimensional feature learning. While these multi-branch designs are promising, a systematic integration that guarantees end-to-end equivariance for structured CAD data is missing.

**Summary and Positioning.** In summary, while existing methods have made significant strides in 3D shape analysis and CAD processing (Guo et al., 2020; Jayaraman et al., 2021a;b), they operate in distinct lanes: CAD-specific GNNs often overlook foundational 3D symmetries, while equivariant networks have not been adapted to the structured complexity of B-reps. We introduce the first framework that provides a rigorous group-theoretic formulation for achieving $SO(3)/O(3)$-equivariance on B-rep graphs, thereby unifying the topological expressivity of GNNs with the robustness guarantees of equivariant deep learning.

## 3 METHODOLOGY

### 3.1 OVERALL FRAMEWORK

We present EquiCAD, a novel framework that systematically integrates $SO(3)/O(3)$-equivariant learning with graph-based reasoning (Gilmer et al., 2017; Wang et al., 2019) for robust 3D solid classification. Our key insight is that CAD models exhibit inherent geometric symmetries that should be preserved throughout the learning process, while simultaneously leveraging the structured topological information embedded in boundary representations (Wu et al., 2021; Jayaraman et al.,

2021b). As illustrated in Figure 1, EquiCAD comprises three core components: data representation, symmetry-aware feature fusion, and topology-aware graph aggregation.

**Data Preprocessing and Representation.** Each 3D solid is represented as a face-adjacency graph $\mathcal{G} = (\mathcal{V}, \mathcal{E})$, where nodes $\mathcal{V}$ correspond to parametric surfaces and edges $\mathcal{E}$ represent shared boundary curves. This representation naturally captures both geometric properties—through surface and curve attributes—and topological relationships via graph connectivity. Each surface $s \in \mathcal{V}$ is uniformly sampled on a regular UV grid, producing a tensor of shape $(B_{\text{node}}, 7, H, W)$ containing 3D coordinates, normal vectors, and trimming masks. Similarly, curves $c \in \mathcal{E}$ are sampled to yield $(B_{\text{edge}}, 6, L)$ tensors encoding coordinates and tangents. All geometric data are normalized via bounding box normalization to ensure invariance to translation and scale.

**Symmetry-Aware Feature Encoding** The core innovation of EquiCAD lies in its dual-path encoding strategy. The *equivariant branch* processes geometric features using $SO(3)/O(3)$-equivariant layers that explicitly preserve transformation properties under 3D rotations and reflections. In parallel, the *CNN branch* employs conventional convolutional networks to capture intricate local patterns in the UV parameter space. This complementary design enables the model to benefit from both symmetry-aware feature extraction and high-capacity local representation learning.

**Topology-aware Graph Aggregation.** Encoded surface and curve features from both branches are fused via element-wise summation and then propagated through a message-passing graph neural network. This graph aggregation module models interactions between adjacent faces and curves, effectively capturing the complex topological relationships that characterize solid models. The final graph-level embedding is obtained through a multiscale readout mechanism that integrates information across all message-passing layers.

## 3.2 SYMMETRY-AWARE FEATURE ENCODING

The equivariant branch forms the mathematical foundation of our approach, ensuring that learned representations transform consistently with the input geometry under 3D rotations and reflections. We adopt similar pathway to extract features of the surface and the curves. Below, we take the surface $s \in \mathcal{V}$ as an example to show the feature encoding method.

**Physical Parity and Irrep Typing.** A fundamental contribution of EquiCAD is our principal group-theoretic formulation for the encoding of B-rep features. Unlike previous works that treat geometric attributes as generic feature vectors, we rigorously assign each attribute type to its corresponding irreducible representation based on transformation properties in the $SO(3)/O(3)$ group.

The mathematical justification comes from the transformation behavior of differential geometric entities under orthogonal transformations. For a rotation/reflection $g \in SO(3)/O(3)$, the transformation rules follow the physical parity of each geometric entity:

- **Curve Encoding ("$2 \times 1o$"):** Both coordinates and tangents transform as polar vectors ($l = 1, p = o$), since for any $g \in SO(3)/O(3)$:

$$g \cdot \mathbf{x} = \rho_1^o(g)\mathbf{x}, \quad g \cdot \mathbf{t} = \rho_1^o(g)\mathbf{t} \tag{1}$$

  where $\mathbf{x}$ denotes the position, $\mathbf{t}$ the tangent vector, and $\rho_l^p(g)$ is the group representation acting on type $(l, p)$ features.

- **Surface Encoding ("$1 \times 1o + 1 \times 1e + 1 \times 0e$"):** We carefully distinguish between different transformation behaviors:

$$\text{Coordinates: } g \cdot \mathbf{x} = \rho_1^o(g)\mathbf{x} \quad \text{(polar vector)} \tag{2}$$
$$\text{Normals: } g \cdot \mathbf{n} = \det(g)\rho_1^e(g)\mathbf{n} \quad \text{(axial vector)} \tag{3}$$
$$\text{Trimming mask: } g \cdot m = \rho_0^e(g)m \quad \text{(true scalar)} \tag{4}$$

  where $\rho_1^e(g)$ is the representation on axial vectors, $\rho_0^e(g)$ is the trivial representation on scalars (identity action), and $m$ is a binary trimming mask.

This careful typing is crucial to maintain physical consistency. The determinant factor $\det(g)$ in the normal transformation explains why normals must be assigned even parity ($p = e$)—they behave as

pseudo-vectors that do not change sign under reflections. Incorrectly assigning normals as polar vectors would violate this fundamental geometric property. More the detailed theoretical foundations of $SO(3)/O(3)$-equivariant deep learning are elaborated in Appendix A.

**Mathematical Foundation.** The mapping from Cartesian coordinates to irreducible representations follows the spherical harmonic expansion. For an input vector field $v(\mathbf{r})$, we compute its projection onto spherical harmonics:

$$a_{li} = \int v(\mathbf{r}) Y_l^i(\theta, \phi) d\Omega, \tag{5}$$

where $v(\mathbf{r})$ is the input vector field, $Y_l^i$ are real spherical harmonics, and $d\Omega$ is the surface element on the unit sphere. The tensor product operation, fundamental to our equivariant layers, follows the Clebsch-Gordan decomposition.

$$(l_1, p_1) \otimes (l_2, p_2) = \bigoplus_{L=|l_1-l_2|}^{l_1+l_2} (L, p_1 \cdot p_2). \tag{6}$$

where $p_1 \cdot p_2$ denotes the parity product ($o \cdot o = e$, $o \cdot e = o$, $e \cdot e = e$). For our curve encoding, the tensor product of two polar vectors yields the following.

$$(1, o) \otimes (1, o) = (0, e) \oplus (1, e) \oplus (2, e). \tag{7}$$

The above equation describes how vector inputs naturally produce scalar and higher-order tensor components. This mathematical foundation ensures that all feature transformations respect the $SO(3)/O(3)$ symmetry group.

**Equivariant Network Architecture.** Our architecture guarantees $SO(3)/O(3)$-equivariance through tensor product operations that respect the Clebsch-Gordan decomposition. The fundamental operation is the equivariant linear layer between the input type $\tau_{\text{in}}$ and the output type $\tau_{\text{out}}$:

$$y = \text{Linear}_{\tau_{\text{in}} \to \tau_{\text{out}}}(s) = \bigoplus_j W_j s^{(j)} \tag{8}$$

where $s \in \mathcal{V}$ is the surface features and $s^{(j)}$ is the component of input features in the $j$-th irrep subspace. The tensor product layers further enrich feature interactions while preserving equivariance.

$$\rho_{(L,p)}(g)(s \otimes y) = (\rho_{(l_1,p_1)}(g)x) \otimes (\rho_{(l_2,p_2)}(g)y) \tag{9}$$

where $\rho_{(L,p)}(g)$ is the action of $g$ on the $(L, p)$ irrep space.

A critical design choice is that the final equivariant layer produces only scalar ($0e$) output, enabling spatial pooling to obtain $SO(3)/O(3)$-invariant embeddings:

$$h_{\text{enn,s}} = \frac{1}{N} \sum_{i=1}^{N} y_i, \tag{10}$$

where $y_i$ is the scalar output at spatial location $i$, and $N$ is the number of sampling points. $h_{\text{enn,s}}$ is the $SO(3)/O(3)$-invariant embeddings for the surface. Similarly, we can follow the above way using different neural weights to obtain the corresponding $SO(3)/O(3)$-invariant embeddings for the curves, i.e., $h_{\text{enn,c}}$.

### 3.3 MULTI-BRANCH FEATURE FUSION

While the equivariant branch provides mathematical guarantees about symmetry preservation, we complement it with conventional CNNs to capture potentially useful patterns that may not fit neatly into the irrep formalism.

**Complementary Encoding Strategies.** The CNN branch processes the same input data through standard convolutional layers, capturing local correlations and patterns that may be difficult to represent in the constrained parameter space of equivariant layers. This branch serves as a high-capacity feature extractor that can learn data-driven representations without explicit symmetry constraints. Finally, we obtain the spatial encodings $h_{s,\text{cnn}}$ and $h_{c,\text{cnn}}$ for the surface $s \in \mathcal{V}$ and curve $c \in \mathcal{E}$, respectively.

**Feature Fusion Strategy.** We employ a simple yet effective summation-based fusion:

$$H_s = \{h_s \| s \in \mathcal{V}\} = h_{s,\text{cnn}} + h_{s,\text{enn}}, \tag{11}$$

$$E = \{e_{uv} \| u, v \in \mathcal{V}\} = h_{c,\text{cnn}} + h_{c,\text{enn}}, \tag{12}$$

where $H_c$ and $H_s$ represent the final curve and surface embeddings, respectively. The two surfaces connected to curve $c$ are $u$ and $v$. This approach allows the model to leverage the strengths of both encoding strategies: the mathematical rigor of equivariant learning and the representational capacity of conventional CNNs.

The fusion occurs before graph aggregation, ensuring that both symmetry-aware and data-driven features participate in the topological reasoning process. This design enables rich interactions between geometric priors and learned patterns throughout the network.

### 3.4 TOPOLOGY-AWARE GRAPH AGGREGATION

The graph aggregation module models the complex relationships between geometric entities in B-rep models while maintaining the learned symmetry properties.

**Message Passing Formulation.** We initialize node features with surface embeddings $H_s$ and edge features with curve embeddings $H_c$. The message passing follows a multi-layer architecture with residual connections. The edge update is given by:

$$e_{uv}^{(l+1)} = M_E \left( (1 + \epsilon_E) e_{uv}^{(l)} + M_P(h_u^{(l)}) + M_P(h_v^{(l)}) \right), \tag{13}$$

where $M_E$ and $M_P$ are MLPs, $\epsilon_E$ is a learnable scalar, and $h_u^{(l)}, h_v^{(l)}$ are the features of nodes $u, v$ at layer $l$.

The node update employs an edge-conditioned convolution:

$$\tilde{h}_u^{(l+1)} = \sum_{v \in \mathcal{N}(u)} W_{uv} h_v^{(l)}, \quad h_u^{(l+1)} = M_N \left( (1 + \epsilon_N) h_u^{(l)} + \tilde{h}_u^{(l+1)} \right), \tag{14}$$

where $\mathcal{N}(u)$ denotes the neighbor set of node $u$, $W_{uv} \in \mathbb{R}^{d_{\text{in}} \otimes d_{\text{out}}}$ is generated from edge features, and $d_{\text{in}}, d_{\text{out}}$ are the input/output dimensions.

**Multi-Scale Graph Readout.** After $L$ message-passing layers, we compute the graph-level embedding using jumping-knowledge aggregation:

$$H_G = \sum_{k=0}^{L} \text{Dropout} \left( \text{Linear}_k \left( \text{POOL}(h^{(k)}) \right) \right), \tag{15}$$

where $h^{(k)}$ is the node feature at layer $k$, and POOL is a permutation-invariant pooling operation (e.g., mean or max). The resulting graph embedding maintains $SO(3)/O(3)$-invariance through our deliberate design of equivariant processing and invariant readout, while also being permutation-invariant to handle arbitrary face orderings. Finally, we adopt cross-entropy loss to optimize the whole neural network.

## 4 EXPERIMENTS

### 4.1 DATASETS

We evaluate EquiCAD on three benchmarks: SolidLetters, Parts, and Features. **SolidLetters** contains 25,000 3D letter shapes with rotational symmetries, serving as a controlled testbed for symmetry-aware methods. **Parts** comprises 10,834 industrial mechanical components with complex topological structures, representing real-world CAD challenges. The Features dataset is newly constructed focusing on fine-grained geometric feature recognition (Zhang et al., 2024).

**Features Dataset Construction:** We construct a novel benchmark from a segmentation-style B-rep corpus by repurposing it into 24 binary classification tasks. Each sub-dataset targets a specific geometric feature type (e.g., slots, pockets, chamfers). For each feature type $c$, we extract positive

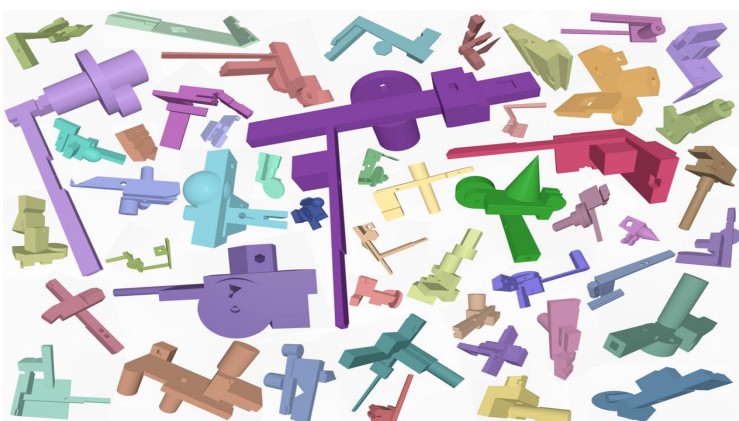

Figure 2: Representative examples from the Features dataset, showcasing diverse geometric features including through slots, blind steps, pockets, and chamfers.

samples from shapes containing annotated regions of type $c$ and select negative samples from shapes completely lacking that feature. We then rank all shapes by geometric-topological complexity using multiple metrics and select the 2,000 most complex positives and 2,000 most complex negatives for the final dataset. This design emphasizes challenging instances and enables focused evaluation on fine-grained geometric attributes. Figure 2 shows representative examples.

**Data Preprocessing:** All models undergo consistent preprocessing: we parse B-rep structures, reconstruct face-edge graphs, and apply bounding-box normalization. Faces are sampled on 16x16 UV grids (7 channels: coordinates, normals, trimming mask), while edges are uniformly sampled into 64-point curves (6 channels: coordinates, tangents).

## 4.2 BASELINE METHODS

We benchmark against two families of methods to cover both unstructured and CAD-structured settings. Point-based baselines include PointNet (Qi et al., 2017a), PointNet++ (Qi et al., 2017b), and DGCNN (Wang et al., 2019): PointNet provides permutation-invariant set processing via symmetric pooling; PointNet++ augments this with hierarchical set abstraction to better capture local neighborhoods; DGCNN forms dynamic $k$NN graphs and applies edge convolutions with neighborhood recomputation for stronger local geometry modeling. CAD-oriented baselines operate directly on B-reps: UV-Net (Jayaraman et al., 2021a) combines UV-domain CNN encoders with a face–edge GNN to jointly model parameterized surfaces and topological adjacencies; AAGNet (Wu et al., 2024) leverages attributed adjacency to propagate geometric/topological cues on the B-rep graph; DTG-BrepGen (Li et al., 2025) incorporates topology-aware constraints and geometric signals beyond UV-only designs. This spectrum establishes a balanced comparison against our symmetry-aware B-rep classifier.

For protocol and fairness, we use the train/val/test splits defined in Section 5.3 and adopt a shared training setup (optimizer, budget, early stopping/model selection on validation accuracy) as detailed therein. CAD methods (UV-Net, AAGNet, DTGBrepGen, ours) receive B-rep UV tensors and face–edge graphs produced by the same preprocessing pipeline; point-based methods (PointNet/PointNet++/DGCNN) are fed uniformly sampled points from the same shapes with identical normalization and, where applicable, shared augmentation. Results are reported as the mean over three random seeds unless stated otherwise, and we use authors' recommended hyperparameters when they conflict with our defaults.

## 4.3 IMPLEMENTATION DETAILS

**Experimental Setup:** We employ fixed dataset splits: 16:4:5 for SolidLetters, 8:1:1 for Parts and Features. For Features, splits are applied independently per sub-dataset. All experiments use Adam optimizer with learning rate 0.001, batch size 32, and are conducted on NVIDIA RTX 3090 GPUs.

| Method | SolidLetters | Parts |
|---|---|---|
| UV-Net (Jayaraman et al., 2021a) | 96.96% | 86.70% |
| PointNet (Qi et al., 2017a) | 97.01% | 81.55% |
| PointNet++ (Qi et al., 2017b) | 97.36% | 88.51% |
| DGCNN (Wang et al., 2019) | 94.76% | 74.11% |
| AAGNet (Wu et al., 2024) | 95.13% | 90.54% |
| DTGBrepGen (Li et al., 2025) | 95.34% | 90.46% |
| EquiCAD | **97.64%** | **91.66%** |

Table 1: Classification accuracy on SolidLetters and Parts datasets.

| Feature Type | PointNet | PointNet++ | DGCNN | UV-Net | AAGNet | DTGBrepGen | **EquiCAD** |
|---|---|---|---|---|---|---|---|
| Rectangular through slot | 0.538 | 0.570 | 0.565 | 0.615 | 0.513 | 0.510 | **0.706** |
| Triangular through slot | 0.542 | 0.567 | 0.524 | 0.858 | 0.537 | 0.794 | **0.982** |
| Circular through slot | 0.569 | 0.580 | 0.503 | 0.510 | 0.522 | 0.470 | **0.594** |
| Triangular passage | 0.537 | 0.535 | 0.483 | 0.859 | 0.591 | 0.515 | **0.998** |
| Rectangular through step | **0.589** | 0.571 | 0.524 | 0.536 | 0.556 | 0.506 | 0.583 |
| 2-Sided through step | 0.541 | 0.572 | 0.548 | 0.512 | 0.546 | 0.513 | **0.594** |
| Slanted through step | 0.553 | 0.570 | **0.588** | 0.518 | 0.546 | 0.479 | 0.579 |
| Rectangular blind step | 0.544 | 0.583 | 0.599 | 0.570 | 0.494 | 0.512 | **0.843** |
| Triangular blind step | 0.561 | 0.570 | 0.547 | 0.823 | 0.521 | 0.537 | **0.979** |
| Circular blind step | 0.543 | 0.593 | 0.525 | 0.557 | 0.501 | 0.510 | **0.604** |
| Rectangular blind slot | 0.541 | 0.514 | 0.525 | 0.495 | 0.520 | 0.527 | **0.572** |
| Horizontal circular end blind slot | 0.517 | 0.570 | 0.554 | 0.509 | 0.531 | 0.509 | **0.570** |
| Vertical circular end blind slot | 0.561 | 0.581 | 0.529 | 0.549 | 0.503 | 0.517 | **0.590** |
| Rectangular pocket | 0.562 | 0.573 | 0.544 | 0.839 | 0.525 | 0.510 | **1.000** |
| Chamfer | 0.518 | 0.519 | 0.559 | 0.575 | 0.518 | 0.511 | **0.615** |
| Illet | 0.568 | **0.598** | 0.582 | 0.476 | 0.551 | 0.511 | 0.550 |
| Average | 0.549 | 0.567 | 0.544 | 0.613 | 0.530 | 0.527 | **0.710** |

Table 2: AUROC performance on 16 challenging feature recognition tasks (mean over 3 runs). Best results in bold.

**EquiCAD Architecture:** Our framework combines $SO(3)/O(3)$-equivariant and conventional CNN branches with follwoing settings. **Equivariant Branch:** Curve encoder: "$2 \times 1o \rightarrow 8 \times 0e + 8 \times 2e \rightarrow 64 \times 0e$"; Surface encoder: "$1 \times 1o + 1 \times 1e + 1 \times 0e \rightarrow 8 \times 0e + 8 \times 1o + 8 \times 1e + 4 \times 2e \rightarrow 64 \times 0e$". **CNN Branch:** 1D CNN for curves ($6 \rightarrow 64 \rightarrow 128 \rightarrow 256$), 2D CNN for surfaces ($7 \rightarrow 64 \rightarrow 128 \rightarrow 256$). **Graph Aggregation:** 3-layer GNN with 64-dimensional hidden states. Models are trained for exactly 100 epochs without early stopping. For SolidLetters and Parts, we report single-run accuracy. For Features dataset, we report mean AUROC over 3 random seeds to ensure statistical reliability.

## 4.4 MAIN RESULTS

**SolidLetters and Parts Datasets:** Table 1 shows classification accuracy across methods. On SolidLetters, most baselines achieve strong performance, yet EquiCAD provides a consistent improvement (97.64%). The Parts dataset reveals more substantial differences: while the best baseline reaches 90.54%, EquiCAD achieves 91.66%—demonstrating superior performance on complex industrial components.

**Features Dataset Analysis:** Table 2 presents AUROC scores for 16 challenging feature recognition tasks (mean over 3 runs). EquiCAD achieves the highest performance in the majority of tasks, with particularly strong gains on geometrically complex features like triangular passages (0.998 vs 0.859) and rectangular pockets (1.000 vs 0.839). This demonstrates a superior fine-grained geometric understanding.

| Configuration | SolidLetters | Parts |
|---|---|---|
| CNN-only | 96.96% | 86.70% |
| Ours-Curve_ENN | 95.57% | 90.52% |
| Ours-Surface_ENN | 95.40% | 90.26% |
| EquiCAD (full) | **97.64%** | **91.66%** |

Table 3: Ablation study.

| Curve Irreps | Surface Irreps | SolidLetters | Parts |
|---|---|---|---|
| 2x1o | 1x1o+1x1e+1x0e (Ours) | **97.64%** | **91.66%** |
| 2x1o | 2x1o+1x0e | 42.49% | 59.37% |
| 2x1o | 1x1o+4x0e | 96.16% | 91.22% |
| 2x1o | 7x0e | 94.96% | 91.19% |
| 1x1o+3x0e | 2x1o+1x0e | 95.53% | 91.09% |
| 1x1o+3x0e | 1x1o+4x0e | 95.67% | 88.46% |
| 1x1o+3x0e | 7x0e | 94.84% | 91.19% |
| 6x0e | 2x1o+1x0e | 96.89% | 88.70% |
| 6x0e | 1x1o+4x0e | 96.87% | 90.48% |
| 6x0e | 7x0e | 96.89% | 90.98% |

Table 4: Robustness analysis: performance under different irrep configurations.

## 4.5 ABLATION STUDY

We systematically evaluate the contribution of the components through a comprehensive ablation design (Table 3) about four variants. **CNN-only:** Remove equivariant encoders from both branches (only CNN components). **Ours-Curve_ENN:** Curve branch uses only CNN; Surface branch uses equivariant encoder + CNN auxiliary. **Ours-Surface_ENN:** Surface branch uses only CNN; Curve branch uses equivariant encoder + CNN auxiliary. **EquiCAD (full):** Both branches use equivariant encoder + CNN auxiliary (our complete model). As shown in Table 3, the full model achieves the best performance on both datasets. Removing equivariant encoding from either branch causes performance degradation, while removing both yields the largest drop. This shows that symmetry-aware features from both curves and surfaces provide complementary benefits.

## 4.6 ROBUSTNESS ANALYSIS

Table 4 evaluates sensitivity to irreducible representation choices. Our physically-motivated configuration (curve: 2x1o; surface: 1x1o+1x1e+1x0e) achieves optimal performance. Critically, mistyping surface normals as polar vectors (2x1o+1x0e) causes catastrophic performance degradation (42.49% on SolidLetters), validating our physical parity assignment. Reducing angular capacity also degrades performance, confirming the importance of directional information. More ablation results are provided in Appendix A.3.

## 5 CONCLUSION

We presented EquiCAD, a novel framework that integrates $SO(3)/O(3)$-equivariant learning with graph neural networks for 3D solid classification. Our key contribution is a symmetry-aware encoding scheme that decomposes B-rep entities into irreducible representations based on physical parity, ensuring rigorous equivariance while capturing geometric-topological relationships. Experiments across SolidLetters, Parts, and our new Features benchmark demonstrate state-of-the-art performance, particularly on complex industrial shapes and fine-grained feature recognition tasks. The consistent superiority of EquiCAD validates the importance of incorporating geometric symmetries for robust CAD analysis. Future work will explore applications to other CAD tasks and more complex equivariant architectures.

## ETHICS STATEMENT

All authors have read and adhered to the ICLR Code of Ethics. This research is foundational and methodological in nature, focusing on advancing geometric deep learning and 3D shape understanding for computer-aided design (CAD) data. Our work exclusively utilizes publicly available CAD benchmark datasets (e.g., SolidLetters, Parts, and the Features dataset), which do not contain personally identifiable information or sensitive data, and no human subjects were involved in this study. Furthermore, our work does not present any other ethical violations.

## REPRODUCIBILITY STATEMENT

To ensure the reproducibility of our work, we have provided comprehensive details throughout the paper and its appendix. All experiments are conducted on publicly available and cited CAD benchmark datasets, including SolidLetters, Parts, and Features. Our proposed EquiCAD framework and its components are thoroughly described in Section 3. Specific implementation details, including hyperparameters, model configurations, and training procedures, are provided in the Section 4.3. Foundational theoretical analysis is also provided in the appendix. The source code, data preprocessing scripts, and the newly constructed Features dataset for our experiments will be made publicly available upon the paper's acceptance to facilitate further research and verification.

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

## A APPENDIX

### A.1 STATEMENT ON THE USE OF LARGE LANGUAGE MODELS

During the preparation of this manuscript, we used the Large Language Model (LLM) to polish the language and correct grammatical errors to improve readability. The LLM was not involved in any core research aspects of the paper, such as research ideation, experimental design, or analysis of results.

### A.2 PRELIMINARIES: $SO(3)/O(3)$-EQUIVARIANT DEEP LEARNING

To ensure our framework is accessible to readers unfamiliar with geometric deep learning, we briefly review the essential concepts of $SO(3)/O(3)$-equivariant neural networks. Our focus is on the orthogonal group $SO(3)/O(3)$—the group of 3D rotations and reflections—which provides the mathematical foundation for handling the symmetries inherent in CAD models.

**$SO(3)/O(3)$-Equivariance Principle.** A function $f : X \to Y$ is *equivariant* to a group $G$ if transforming the input by any group element $g \in G$ results in a predictable transformation of the output:

$$f(\rho_X(g) \cdot x) = \rho_Y(g) \cdot f(x), \quad \forall g \in G, \tag{16}$$

where $\rho_X$ and $\rho_Y$ are representations of $G$ on the input and output spaces, respectively. In our context, $G = SO(3)/O(3)$, and equivariance ensures that rotating or reflecting a B-rep model consistently transforms its learned features—a crucial property for robust CAD analysis.

**Irreducible Representations of $SO(3)/O(3)$.** To build $SO(3)/O(3)$-equivariant networks, we decompose features into *irreducible representations* (irreps)—the fundamental building blocks that transform in a simple, predictable way under group actions. Each irrep is characterized by an integer *degree $l \geq 0$* and a *parity $p \in \{e, o\}$*:

- Degree $l$ determines the dimensionality $(2l+1)$ and angular frequency of the transformation behavior.
- Parity $p$ specifies behavior under reflection: *even* ($p = e$) features remain unchanged, while *odd* ($p = o$) features change sign.

For example, scalar values ($l = 0$, $p = e$) are invariant to rotations, while polar vectors (e.g., coordinates) transform as $l = 1$, $p = o$ irreps.

**Feature Encoding with Multiplicities.** Practical $SO(3)/O(3)$-equivariant networks compose features as direct sums of irreps. For each irrep type $(l, p)$, we can include multiple independent copies (multiplicity $m$). Thus, a feature vector containing $K$ distinct irrep types has total dimension:

$$D = \sum_{i=1}^{K} m_i \cdot (2l_i + 1). \tag{17}$$

Following conventions in libraries like e3nn (Geiger et al., 2022), we notationally express feature types as concatenations like "$2 \times 1o + 3 \times 0e$", denoting two vector-valued ($l = 1$, odd) and three scalar ($l = 0$, even) channels.

**$SO(3)/O(3)$-Equivariant Linear Maps and Nonlinearities.** $SO(3)/O(3)$-Equivariant linear layers between input and output features of types $\rho_{\text{in}}$ and $\rho_{\text{out}}$ must satisfy the intertwining condition:

$$W \cdot \rho_{\text{in}}(g) = \rho_{\text{out}}(g) \cdot W, \quad \forall g \in SO(3)/O(3). \tag{18}$$

Such constraints drastically reduce the number of free parameters in $W$, improving generalization. For nonlinearities, standard activation functions (e.g., ReLU) generally break equivariance. Instead, we use *gate nonlinearities* (Weiler et al., 2018) that combine tensor products of irreps with learned gating scalars, preserving equivariance while enabling complex feature learning.

These foundations enable our key contribution: adapting $SO(3)/O(3)$-equivariance to the structured graph domain of CAD B-reps, building upon established graph neural network principles (Gilmer et al., 2017; Kipf & Welling, 2017) while incorporating geometric symmetry constraints.

### A.3 COMPLETE ROBUSTNESS EXPERIMENTS

In this subsection, we present a comprehensive robustness analysis table that combines results shown in the main paper with additional configurations not included due to space constraints. This extended table provides a full view of the performance under various irreducible representation choices and physical parity assignments.

| Curve Irreps | Surface Irreps | SolidLetters | Parts |
|---|---|---|---|
| 2x1o | 1x1o+1x1e+1x0e (Ours) | **97.64%** | **91.66%** |
| 2x1o | 2x1o+1x0e | 42.49% | 59.37% |
| 2x1o | 1x1o+4x0e | 96.16% | 91.22% |
| 2x1o | 7x0e | 94.96% | 91.19% |
| 1x1o+3x0e | 2x1o+1x0e | 95.53% | 91.09% |
| 1x1o+3x0e | 1x1o+4x0e | 95.67% | 88.46% |
| 1x1o+3x0e | 7x0e | 94.84% | 91.19% |
| 6x0e | 2x1o+1x0e | 96.89% | 88.70% |
| 6x0e | 1x1o+4x0e | 96.87% | 90.48% |
| 6x0e | 7x0e | 96.89% | 90.98% |
| 1X1o+3x0e | 1x1o+1x1e+1x0e | 95.30% | 90.63% |
| 6x0e | 1x1o+1x1e+1x0e | 95.37% | 90.89% |
| 2x1o | 2x1e+1x0e | 95.43% | 91.05% |
| 1x1o+3x0e | 2x1e+1x0e | 95.48% | 90.46% |
| 6x0e | 2x1e+1x0e | 95.16% | 90.59% |

Table 5: Comprehensive robustness analysis: combining main paper results with additional configurations.

