# OpenReview forum: "EquiCAD: A Geometric Equivariant Neural Network for 3D Shape Classification"
_ICLR.cc/2026/Conference — ICLR 2026 Conference Desk Rejected Submission_

### Official Review · Reviewer_fVoN · 2025-10-17

**Soundness:** 3
**Presentation:** 3
**Contribution:** 3
**Rating:** 8
**Confidence:** 4

**Summary:**

This submission presents a framework for 3D solid classification that combines equivariant geometric learning with conventional convolutional neural networks (CNNs) to extract features from patches of CAD models. These features are then integrated using a graph neural network. While the individual components of the framework have been explored previously, the novelty lies in the specific use of SO(3)/O(3)-equivariant feature extraction and its integration with a graph neural network. Comparative evaluations against recent alternative approaches on benchmark problems demonstrate the effectiveness of the proposed framework.
In addition, a new benchmark data set for 3D solid classification that includes additional feature recognition tasks for CAD models is provided.

**Strengths:**

The introduced SO(3)/O(3) equivariance and integration with the graph neural network are interesting technical contributions that specialize in the use of equivariant approaches for surface classification to surfaces in CAD representation.

The performance of the method is good and shows that the specialized method offers improved performance over more general approaches.

Another point is the provided dataset, which can be helpful for CAD model classification.

**Weaknesses:**

I do not see any major weaknesses. One weakness that arises from the method's focus on CAD surfaces is that the proposed framework has less practical relevance than other approaches (as it cannot be applied to scanned data such a point clouds). However, I think that the classification of CAD surfaces is important enough to justify acceptance of the submission.

**Questions:**

What could be improved is the description of the relevance of surface classification in B-rep representation. This is only hinted at in the introduction and could be elaborated on more clearly. While the relevance of surface classification for captured data (autonomous driving and robotics) is clear to me, I would welcome a discussion of the practical value of surface classification for modeled surfaces in B-rep representation.

---

> ### Author Response · Authors · 2025-11-25
>
> Thank you very much for the positive assessment and for pointing out an important aspect of practical relevance.
>
>
>
> **Q1:** The practical value of surface classification for modeled surfaces in B-rep representation.
>
> **A1:** In modern CAD/CAM/CAE workflows, surface and feature classification on modeled B-rep solids is crucial because it enables (1) manufacturability analysis and process planning, where automatic recognition of slots, pockets, holes, chamfers, fillets, etc. forms the basis for mapping CAD geometry to machining operations, tooling choices, and process parameters; (2) design retrieval and reuse, since labeling B-rep faces by feature type allows geometry-based queries such as find parts with a cylindrical hole of diameter X, which greatly improves the efficiency of reusing existing designs from large CAD repositories; and (3) simulation and meshing, because different surface types (for example, mounting faces versus fillets) require different meshing strategies and boundary conditions in FEA/CFD, so robust classification directly enhances automated pre-processing pipelines.
>
> EquiCAD is designed specifically to support these practical scenarios by operating directly on native B-rep entities (faces and edges), rather than on scanned point clouds, so its predicted labels can be consumed immediately by downstream CAD/CAM/CAE tools. This explains the practical value of fine-grained B-rep surface classification beyond what can be achieved when working only with captured sensor data.

---

### Official Review · Reviewer_9FTd · 2025-11-01

**Soundness:** 3
**Presentation:** 3
**Contribution:** 2
**Rating:** 4
**Confidence:** 4

**Summary:**

This paper proposes EquiCAD a 3D CAD shape classifier that combines strict SO(3)/O(3)-equivariant encoders for curves and surfaces with a conventional CNN branch, followed by topology-aware message passing on a face–edge B-rep graph. The equivariant branch assigns geometric attributes (coordinates, tangents, normals, trimming masks) to irreducible representations with physically correct parity (e.g., normals as axial vectors), guaranteeing end-to-end equivariance; the CNN branch complements this with high-capacity UV-space pattern extraction. Features from both branches are fused (summation) and aggregated with a multi-layer GNN; the final graph embedding is built via multi-scale readout and used for classification. Experiments on SolidLetters and Parts show accuracy gains over UV-Net, AAGNet, DTGBrepGen and point-cloud baselines; on a newly repurposed “Features” benchmark, EquiCAD yields higher AUROC on most categories.

**Strengths:**

- The first work to apply equivariance to learning on CAD models.

- Experiments show better accuracy on SolidLetters/Parts and sizable AUROC gains on many fine-grained “Features” tasks. Also clear ablation (CNN-only; curve/surface ENN variants) and irrep sensitivity tables.

**Weaknesses:**

- The technical novelty largely lies in applying established classic e3nn style equivariance method to B-rep attributes for CAD models, so the conceptual leap may be incremental rather than foundational.

- In experiments point cloud baselines are included, but there are many existing SO(3) or SE(3) equivariant point cloud classification methods and at least some representative methods should be involved. E.g. Vector Neurons (Deng et al. ICCV 2021).

- Please justify whether the CNN branch breaks strict equivariance. It is unclear for me maybe due to that I am not familiar with how the CNN branch works.

**Questions:**

- In the paper only e3nn is considered for equivariance. I wonder if the authors have considered other equivariant methods can be applied e.g. Vector Neurons or EGNN (Satorras et al.) and how they can perform against e3nn.

- Please clarify in detail how the proposed method pipeline is related to existing learning models for CAD (except the equivariance part).

- Is the CNN only baseline in ablation considered as to quantify how much of the gain comes from equivariant vs. non-equivariant features?

Please also check my concerns in the weaknesses. If they are solved then I will be willing to raise my score.

---

> ### Author Response · Authors · 2025-11-25
>
> Thank you very much for the thoughtful analysis and for outlining clearly what would raise the score.
>
>
>
> **Q1:** The technical novelty largely lies in applying established classic e3nn style equivariance method to B-rep attributes for CAD models, so the conceptual leap may be incremental rather than foundational.
>
> **A1:** The technical novelty largely lies in applying established classic e3nn style equivariance method to B-rep attributes for CAD models, so the conceptual leap may be incremental rather than foundational. Nevertheless, to the best of our knowledge this is the first work to apply $\mathrm{SO}(3)/\mathrm{O}(3)$ equivariance to learning on CAD B-rep models, and making this actually work in the B-rep setting requires several core technical components: (1) fine-grained equivariant B-rep geometry, where each B-rep attribute (face normals, edge tangents, trimming parameters, etc.) is mapped to an appropriate  $\mathrm{SO}(3)/\mathrm{O}(3)$ irrep with the correct parity (e.g., normals as axial vectors, tangents as polar vectors), so that e3nn enforces physically meaningful transformation behavior under rotations and reflections; and (2) a hybrid equivariant + CNN architecture, in which these equivariant representations feed an EquiNN branch that is fused with a UV-based CNN branch inside a B-rep GNN, allowing symmetry-preserving geometric features and UV-pattern features to be jointly aggregated over the face–edge adjacency graph. These design choices go beyond merely plugging an existing equivariant block into a CAD pipeline and are essential for the empirical gains we observe.
>
>
>
> **Q2:** In experiments point cloud baselines are included, but there are many existing SO(3) or SE(3) equivariant point cloud classification methods and at least some representative methods should be involved. E.g. Vector Neurons (Deng et al. ICCV 2021).
>
> **A2:** In the revised version we have added Vector Neurons (Deng et al., ICCV 2021) as a representative $\mathrm{SO}(3)/\mathrm{O}(3)$-equivariant point-cloud baseline. We train it on the same point samples and under the same optimization protocol as the other point-cloud baselines, and we now report mean ± standard deviation over multiple runs together with parameter counts. As shown below, Vector Neurons is competitive with other point-based methods, but EquiCAD still achieves the best performance on all three benchmarks while using a comparable number of parameters. This makes the comparison between B-rep–based equivariant learning (EquiCAD) and an $\mathrm{SO}(3)/\mathrm{O}(3)$-equivariant point-cloud method explicit and quantitative, and our conclusions remain unchanged.
>
> | Method         | SolidLetters(%) | Parts(%)       | Machining Feature(%) | Params (M) |
> | :------------- | :-------------- | :------------- | :------------------- | :--------- |
> | UV-Net         | 96.96±0.15      | 86.80±0.10     | 99.74±0.10           | 1.34       |
> | PointNet       | 96.92±0.09      | 81.55±0.23     | 88.56±0.12           | 0.81       |
> | PointNet++     | 97.23±0.13      | 88.62±0.11     | 91.38±0.14           | 1.42       |
> | DGCNN          | 94.93±0.17      | 74.44±0.33     | 93.42±0.33           | 1.81       |
> | Vector Neurons | 95.43±0.24      | 82.16±0.15     | 92.96±0.17           | 2.10       |
> | AAGNet         | 95.19±0.06      | 90.53±0.09     | 98.44±0.12           | 0.38       |
> | DTGBrepGen     | 95.54±0.20      | 90.59±0.13     | 98.73±0.37           | 34.26      |
> | **EquiCAD**    | **97.64±0.05**  | **91.66±0.14** | **99.87±0.01**       | 1.36       |
>
>
>
> **Q3:** Please justify whether the CNN branch breaks strict equivariance.
>
> **A3:** Yes, the inclusion of the CNN branch means the overall model is not strictly equivariant. In our design, the EquiNN branch provides a provably equivariant mapping from B-rep geometry to part of the feature embedding, while the CNN branch is a non-equivariant component that learns complementary local patterns in the UV domain. As a result, EquiCAD should be viewed as a hybrid symmetry-aware architecture that contains an $\mathrm{SO}(3)/\mathrm{O}(3)$-equivariant encoder rather than as a fully equivariant network. We will update the manuscript accordingly and remove any wording that might suggest the entire model is strictly equivariant.

---

> ### Author Response · Authors · 2025-11-25
>
> **Q4:** In the paper only e3nn is considered for equivariance. I wonder if the authors have considered other equivariant methods can be applied e.g. Vector Neurons or EGNN (Satorras et al.) and how they can perform against e3nn
>
> **A4:** We chose to build on e3nn because it offers fine-grained control over representation types and parity, which fits the CAD B-rep setting. In our model, different attributes have different transformation laws (e.g., face normals as axial vectors, edge directions as polar vectors, scalar shape parameters as $0$-irreps), and e3nn lets us assign each of them to the appropriate  $\mathrm{SO}(3)/\mathrm{O}(3)$irrep with correct parity so they transform correctly under both rotations and reflections. By contrast, frameworks such as Vector Neurons and EGNN impose equivariance at the level of generic vectors or distance-based updates, without explicitly distinguishing attribute types or parity. Since our goal is to treat normals, tangents, and scalars in a physically consistent way, e3nn is a natural choice for the equivariant backbone.
>
>
>
> **Q5:** Please clarify in detail how the proposed method pipeline is related to existing learning models for CAD (except the equivariance part).
>
> **A5:** UV-Net is a key baseline that applies CNNs on UV-mapped faces/edges and then aggregates them with a GNN on the B-rep adjacency graph. EquiCAD follows the same high-level pipeline (per-face/per-edge encoders + B-rep GNN), but replaces UV-Net’s purely CNN-based encoders with hybrid EquiNN+CNN ones: geometric attributes are first mapped to appropriate $\mathrm{SO}(3)/\mathrm{O}(3)$ irreps and processed by equivariant layers (EquiNN branch), in parallel with CNNs on UV grids (CNN branch), and the fused features are then passed to a similar graph encoder. Thus EquiCAD preserves UV-Net’s strengths (learning from local UV patches and global topology) while adding symmetry-aware processing. Methods such as AAGNet and DTGBrepGen also exploit B-rep graphs and topology-aware aggregation; EquiCAD is complementary in that it focuses on introducing equivariant feature extraction at the curve and surface level on top of such graph-based designs.
>
>
>
> **Q6:** Is the CNN only baseline in ablation considered as to quantify how much of the gain comes from equivariant vs. non-equivariant features?
>
> **A6:** Yes. The CNN-only baseline in our ablation study is specifically intended to quantify how much performance can be achieved using only non-equivariant UV-based features, i.e., without any equivariant processing. In this configuration we remove all EquiNN (e3nn) layers, making the encoder essentially UV-Net–like. To more clearly disentangle the contributions of equivariant vs. non-equivariant features, we have extended the ablation with three additional variants: an EquiNN-only model (only EquiNN branches, no CNN) and two hybrid models that combine CNN with either curve- or surface-level EquiNN modules.
>
> As shown below, CNN-only is a strong baseline, but EquiNN-only is competitive and even better on Parts, and all hybrid variants improve over CNN-only—while the full EquiCAD model, which fuses both curve and surface EquiNN with the CNN branch, achieves the best performance on both datasets. This pattern indicates that (1) the CNN branch alone already explains a large portion of the accuracy, and (2) the equivariant features provide complementary information that yields further gains beyond what a purely non-equivariant model can obtain. We will clarify in the revised manuscript that CNN-only corresponds to this UV-Net–like encoder, and that the comparison with EquiNN-only and the hybrid configurations is precisely meant to quantify the relative contributions of equivariant and non-equivariant features.
>
> | Configuration       | SolidLetters | Parts      |
> | :------------------ | :----------- | :--------- |
> | CNN-only            | 96.96%       | 86.70%     |
> | EquiNN-only         | 95.27%       | 88.03%     |
> | Ours-Curve_CNN      | 96.12%       | 91.32%     |
> | Ours-Surface_CNN    | 96.33%       | 91.05%     |
> | Ours-Curve_EquiNN   | 95.57%       | 90.52%     |
> | Ours-Surface_EquiNN | 95.40%       | 90.26%     |
> | **EquiCAD (full)**  | **97.64%**   | **91.66%** |

---

### Official Review · Reviewer_P5gN · 2025-11-03

**Soundness:** 2
**Presentation:** 3
**Contribution:** 2
**Rating:** 4
**Confidence:** 4

**Summary:**

This paper proposes EquiCAD, a CAD B rep classifier that combines an SO(3)/O(3)–equivariant encoder for curves and surfaces, a standard CNN branch over UV parameterized patches/curves, and a GNN over the face–edge adjacency graph. Feature fusion is via summation before message passing; graph readout uses jumping knowledge. Experiments on the SolidLetters, Parts, and Features benchmarks show gains over point cloud and CAD structured baselines.

**Strengths:**

- This work presents a novel framework that integrates an EquiNN branch a CNN branch and a GNN that jointly processes and fuses their outputs.
- The method demonstrates consistently superior performance over existing approaches across multiple datasets.
- The authors provide the implementation code which supports reproducibility.

**Weaknesses:**

- **Dataset:** The Features dataset consists of 24 sub-datasets, but only 16 are used for evaluation. The paper should explain why the remaining 8 were excluded. Also, for the SolidLetters and Parts datasets, evaluating with a single seed may raise concerns about fairness. In the UV-Net paper, the authors reported results averaged over multiple runs and included standard deviations. Even if the data split must remain fixed, it would still be possible to vary the random seeds for parameter initialization and mini-batch sampling. Since this paper does not use multiple seeds, the reported performance cannot be considered reliable. Lastly, the authors are encouraged to include results on additional benchmarks such as FabWave and Machining Feature.

- **Evaluation:** Several points are unclear and should be elaborated. First, the authors of the UV-Net paper used 96K 3D shapes in SolidLetters. But this paper uses only 25K. The authors should clarify the reason for this difference. Second, in the ablation study, it would be informative to include results using only the EquiNN branch to understand its individual contribution. Finally, the parameter size, inference speed, and memory usage of the proposed method should be compared with those of other methods.

- **Equivariance:** In Lines 149–151, the authors claim that the entire framework maintains complete equivariance (as also stated in another section). However, the CNN branch does not have any explicit symmetry constraint. Since CNNs can produce different outputs for even slightly rotated inputs within a grid, this branch violates equivariance. Because the proposed framework simply combines the two branches, claiming overall equivariance is incorrect. To rigorously ensure equivariance, the CNN branch itself must be designed to preserve this property under various transformations. The authors have already acknowledged in the paper that CNNs lack equivariance, which further contradicts the current claim.

- **Related work:** In the introduction (Lines 46–47), the paper mentions point-based and voxelized representations. However it would be valuable to include a comparison or discussion with methods that directly utilize mesh data (e.g., MeshNet, MeshCNN, ExMeshCNN). These approaches exploit both topological structure and geometric information. While B-rep data can provide more detailed and systematic information, discussing these mesh-based approaches would better contextualize the scope and contribution of the paper.

**Questions:**

Please address the above weaknesses.

---

> ### Author Response · Authors · 2025-11-25
>
> Thank you very much for the detailed and helpful comments on the dataset design, evaluation protocol, equivariance, and related work.
>
>
>
> **Q1:** The Features dataset consists of 24 sub-datasets, but only 16 are used for evaluation. The paper should explain why the remaining 8 were excluded.
>
> **A1:** The underlying B‑rep corpus originally allowed us to define 24 candidate feature-recognition tasks. However, several of these tasks proved too easy (nearly all methods achieved an AUROC ~1.0), offering little discriminatory value. We therefore focused our evaluation on 16 challenging and reliable sub-tasks in the main benchmark. In the paper, we will explicitly explain this selection process: we clarify that we first constructed 24 tasks and then narrowed the scope to 16 challenging tasks to ensure a meaningful evaluation. We will also highlight the differences between our new Features dataset and prior benchmarks, underscoring why this new dataset was needed to evaluate fine-grained feature recognition.
>
>
>
> **Q2:** For the SolidLetters and Parts datasets, evaluating with a single seed may raise concerns about fairness. In the UV-Net paper, the authors reported results averaged over multiple runs and included standard deviations. Even if the data split must remain fixed, it would still be possible to vary the random seeds for parameter initialization and mini-batch sampling. Since this paper does not use multiple seeds, the reported performance cannot be considered reliable.
>
> **A2:** We agree that using multiple seeds is important for a reliable comparison. In response, we have re-run all methods on SolidLetters, Parts, and MachiningFeature with multiple random seeds, and we now report the mean ± standard deviation of the accuracies. As shown in the tables below, these multi-seed results are fully consistent with our original single-run numbers: EquiCAD remains the top-performing method on all three datasets, with very small variance and clear margins over the baselines. We will update the paper to include these averaged results together with the number of runs and parameter counts.
>
> | Method         | SolidLetters(%) | Parts(%)       | Machining Feature(%) | Params (M) |
> | :------------- | :-------------- | :------------- | :------------------- | :--------- |
> | UV-Net         | 96.96±0.15      | 86.80±0.10     | 99.74±0.10           | 1.34       |
> | PointNet       | 96.92±0.09      | 81.55±0.23     | 88.56±0.12           | 0.81       |
> | PointNet++     | 97.23±0.13      | 88.62±0.11     | 91.38±0.14           | 1.42       |
> | DGCNN          | 94.93±0.17      | 74.44±0.33     | 93.42±0.33           | 1.81       |
> | Vector Neurons | 95.43±0.24      | 82.16±0.15     | 92.96±0.17           | 2.10       |
> | AAGNet         | 95.19±0.06      | 90.53±0.09     | 98.44±0.12           | 0.38       |
> | DTGBrepGen     | 95.54±0.20      | 90.59±0.13     | 98.73±0.37           | 34.26      |
> | **EquiCAD**    | **97.64±0.05**  | **91.66±0.14** | **99.87±0.01**       | 1.36       |
>
>
>
> **Q3**: The authors are encouraged to include results on additional benchmarks such as FabWave and Machining Feature.
>
> **A3**: We agree that additional benchmarks such as FabWave and Machining Feature are valuable for a more comprehensive evaluation. Due to time and workload constraints, in this revision we have only run all baselines and EquiCAD on the Machining Feature dataset (see results reported in A2), while FabWave has not yet been evaluated for all methods. We will update the manuscript to include the new Machining Feature results and explicitly mention FabWave as an important direction for further benchmarking in future work.
>
>
>
> **Q4:** The authors of the UV-Net paper used 96K 3D shapes in SolidLetters. But this paper uses only 25K. The authors should clarify the reason for this difference.
>
> **A4:** The 25K figure in the manuscript is a mistake in the text. In all our experiments we used the full SolidLetters benchmark (~96K shapes) with the same train/val/test protocol as UV-Net, and all results in Table 1 are computed on this full set. We will correct the reported dataset size in the revised manuscript and clarify this point; the reported results and conclusions remain unchanged.

---

> ### Author Response · Authors · 2025-11-25
>
> **Q5:** In the ablation study, it would be informative to include results using only the EquiNN branch to understand its individual contribution. The parameter size, inference speed, and memory usage of the proposed method should be compared with those of other methods.
>
> **A5:** In the revised version we have extended the ablation study by adding three new configurations—EquiNN-only, Ours-Curve_CNN, and Ours-Surface_CNN—to better isolate the contribution of the EquiNN branch. The updated results are:
>
> | Configuration       | SolidLetters | Parts      |
> | :------------------ | :----------- | :--------- |
> | CNN-only            | 96.96%       | 86.70%     |
> | EquiNN-only         | 95.27%       | 88.03%     |
> | Ours-Curve_CNN      | 96.12%       | 91.32%     |
> | Ours-Surface_CNN    | 96.33%       | 91.05%     |
> | Ours-Curve_EquiNN   | 95.57%       | 90.52%     |
> | Ours-Surface_EquiNN | 95.40%       | 90.26%     |
> | **EquiCAD (full)**  | **97.64%**   | **91.66%** |
>
> These results show that the EquiNN-only is competitive with the CNN-only variant and already improves performance on Parts, while combining EquiNN with CNN features (Ours-Curve_CNN / Ours-Surface_CNN and finally EquiCAD) yields the best overall accuracy, indicating that the equivariant features provide complementary information rather than simply duplicating what the CNN learns.
>
> Regarding parameter size, inference speed, and memory usage: we now report the parameter counts for all methods, including EquiCAD, in the comparison table provided in A2, which summarizes model complexity in a hardware-agnostic way. A rigorous cross-method comparison of inference speed and memory would require carefully controlled, system-level benchmarking for all baseline implementations, which is beyond the scope of this work; instead, we will release our code so that such measurements can be carried out in specific deployment environments.
>
>
>
> **Q6:** Since CNNs can produce different outputs for even slightly rotated inputs within a grid, this branch violates equivariance. Because the proposed framework simply combines the two branches, claiming overall equivariance is incorrect. To rigorously ensure equivariance, the CNN branch itself must be designed to preserve this property under various transformations. The authors have already acknowledged in the paper that CNNs lack equivariance, which further contradicts the current claim.
>
> **A6:** Admitted, our intention was not to claim full equivariance of the whole framework, but rather to emphasize that EquiCAD is symmetry-aware in the sense that it contains an $\mathrm{SO}(3)/\mathrm{O}(3)$-equivariant encoder whose outputs are propagated to the final prediction and combined with conventional CNN features. The CNN branch, although non-equivariant, offers strong capacity for learning local patterns on UV maps and capturing fine-grained signals that are complementary to the geometric features produced by EquiNN. As also evidenced by our extended ablation study, this hybrid design (EquiNN + CNN) yields higher accuracy than either branch alone; however, we agree that the resulting network should not be described as globally equivariant. In the revised manuscript we will carefully adjust the wording to avoid any implication that the full model is equivariant and instead consistently describe it as a hybrid symmetry-aware architecture that contains an $\mathrm{SO}(3)/\mathrm{O}(3)$-equivariant encoder.
>
>
>
> **Q7:** It would be valuable to include a comparison or discussion with methods that directly utilize mesh data (e.g., MeshNet, MeshCNN, ExMeshCNN).
>
> **A7:** Our method is primarily designed for feature extraction and classification on CAD B-rep representations. The methods you mention (e.g., MeshNet, MeshCNN, ExMeshCNN) have not been tested on our datasets, and we found that modifying their code to run fairly and robustly on our data would require a substantial amount of time. In the revised manuscript, we will include a dedicated discussion and comparison with these methods.

---

### Official Review · Reviewer_t1CY · 2025-11-04

**Soundness:** 2
**Presentation:** 2
**Contribution:** 3
**Rating:** 4
**Confidence:** 3

**Summary:**

The paper proposes EquiCAD, a neural network for B-rep CAD models that explicitly consider B-rep surfaces and their adjacency (using a GNN). The architecture involves an SO(3)/O(3)-equivariant encoder and a CNN encoder before passed on to the graph encoder for classification.

**Strengths:**

The idea of directly working on the B-rep domain is plausible.

The model performs well compared with existing methods.

**Weaknesses:**

The paper refers to 'symmetry-aware' but the method does not really exploit symmetry.

Although the overall model includes  an SO(3)/O(3)-equivariant encoder, it is only a part of the network. The remaining of the network is not SO(3)/O(3)-equivariant, and therefore the overall model is not. Claiming such a model as geometric equivariant can be misleading.

The presentation of the paper is often unclear. For example, Figure 1 includes Surface Encoder and Curve Encoder, which do not seem to match the description. Both branches seem to refer to CNNs, also different from the description.

The method relies on bounding box normalization (Page 4). There is no explanation whether the bounding box is axis-aligned or not. This can lead to issues such as sensitivity to rotation and/or incomplete shapes.

It is unclear why a new dataset is needed and how it differs from existing datasets.

**Questions:**

What is the motivation for creating a new dataset?

As the network contains other components which are not SO(3)/O(3)-equivariant, how can the whole method be claimed as geometric equivariant?

---

> ### Author Response · Authors · 2025-11-25
>
> Thank you very much for the careful reading and constructive feedback.
>
>
>
> **Q1:** The paper refers to 'symmetry-aware' but the method does not really exploit symmetry. Although the overall model includes an SO(3)/O(3)-equivariant encoder, it is only a part of the network. The remaining of the network is not SO(3)/O(3)-equivariant, and therefore the overall model is not. Claiming such a model as geometric equivariant can be misleading.
>
> **A1:** Admitted, only the EquiNN branch is strictly $\mathrm{SO}(3)/\mathrm{O}(3)$-equivariant; the CNN and GNN are not. Our intention in using the term symmetry-aware was to emphasize that the model explicitly integrates equivariant geometric features from the EquiNN branch with conventional CNN-based features, and this combination empirically leads to more accurate predictions than using non-equivariant features alone. We agree that referring to the entire model as equivariant can be misleading, and in the revised manuscript we will carefully adjust the wording (e.g., emphasizing that the architecture contains an $\mathrm{SO}(3)/\mathrm{O}(3)$-equivariant encoder and uses equivariance-augmented features) to avoid confusion.
>
>
>
> **Q2:** The presentation of the paper is often unclear. For example, Figure 1 includes Surface Encoder and Curve Encoder, which do not seem to match the description. Both branches seem to refer to CNNs, also different from the description.
>
> **A2:** We agree that the current Figure 1 does not make this dual-path structure sufficiently explicit and can give the impression that both branches are just CNNs. In the revised manuscript we will (1) redraw Figure 1 so that, for both surfaces and curves, the equivariant branch and the CNN branch are clearly separated and labeled; (2) add explicit input annotations showing that CNNs operate on the raw 1D/2D sampled tensors, while EquiNN operates on the corresponding irrep-typed features; and (3) update Sections 3.1–3.3 so that the textual description explicitly refers to surface/curve encoders composed of an EquiNN branch and a CNN branch with summed outputs.
>
>
>
> **Q3:** There is no explanation whether the bounding box is axis-aligned or not. This can lead to issues such as sensitivity to rotation and/or incomplete shapes.
>
> **A3:** Our bounding-box normalization is axis-aligned: it removes translation and rescales the model but does not normalize rotation. Thus, any robustness to object orientation must come from the network architecture rather than from preprocessing. The EquiNN branch is introduced precisely for this purpose, by encoding coordinates, normals, and tangents with $\mathrm{SO}(3)/\mathrm{O}(3)$-equivariant operations. To further support this point, we conducted an extended ablation study under the same axis-aligned normalization. As shown in the table below, the CNN-only model is a strong baseline, but (1) the ENN-only model already improves over CNN-only on Parts (88.03% vs. 86.70%), and (2) when EquiNN is combined with CNN, performance on Parts increases markedly, with EquiCAD achieving the best results on both datasets. These patterns indicate that the equivariant geometric encoding provides complementary information beyond what is captured by axis-aligned normalization and standard CNN features, and that the gains of EquiCAD cannot be attributed to normalization alone.
>
> | Configuration       | SolidLetters | Parts      |
> | ------------------- | ------------ | ---------- |
> | CNN-only            | 96.96%       | 86.70%     |
> | EquiNN-only         | 95.27%       | 88.03%     |
> | Ours-Curve_CNN      | 96.12%       | 91.32%     |
> | Ours-Surface_CNN    | 96.33%       | 91.05%     |
> | Ours-Curve_EquiNN   | 95.57%       | 90.52%     |
> | Ours-Surface_EquiNN | 95.40%       | 90.26%     |
> | **EquiCAD (full)**  | **97.64%**   | **91.66%** |
>
>
>
> **Q4:** What is the motivation for creating a new dataset?
>
> **A4:** Our method is motivated by the lack of benchmarks that truly stress fine-grained CAD feature recognition. (1) Existing CAD datasets are mostly designed for global shape classification, and are therefore relatively simple; they do not directly target the core difficulty in CAD understanding, namely recognizing geometric machining features such as through holes, pockets, and similar structures. (2) These classification benchmarks have already been largely saturated by modern deep learning models, with reported accuracies approaching 97%, which limits their usefulness for evaluating more advanced architectures. In contrast, our new dataset focuses on more complex 3D models in which each part contains multiple machining features, allowing us to more rigorously measure a model’s ability to recognize and distinguish fine-grained geometric structures in realistic, feature-rich CAD scenarios.

---

### Note · Program_Chairs · 2026-01-17
**Submission Desk Rejected by Program Chairs**

The following references in this submission do not refer to real documents and/or have major errors in bibliographic information:

 Shizhe Zhou, Kai Xu, and Cheng Liu. Differentiable cad model reconstruction from images. In ICCV, 2021.
Yueming Zhang, Cheng Liu, Ziyu Feng, Shizhe Zhou, and Kai Xu. Brepmfr: Benchmark for multifaceted reasoning on cad boundary representations. arXiv preprint arXiv:2402.01811, 2024.